# A Systematic Review of Extreme Phenotype Strategies to Search for Rare Variants in Genetic Studies of Complex Disorders

**DOI:** 10.3390/genes11090987

**Published:** 2020-08-25

**Authors:** Sana Amanat, Teresa Requena, Jose Antonio Lopez-Escamez

**Affiliations:** 1Otology & Neurotology Group CTS495, Department of Genomic Medicine, GENYO—Centre for Genomics and Oncological Research—Pfizer/University of Granada/Junta de Andalucía, PTS, 18016 Granada, Spain; sana.amanat@genyo.es; 2Centre for Discovery Brain Sciences, Edinburgh Medical School: Biomedical Sciences, University of Edinburgh, Edinburgh EH8 9JZ, UK; mrequena@ed.ac.uk; 3Department of Otolaryngology, Instituto de Investigación Biosanitaria ibs.GRANADA, Hospital Universitario Virgen de las Nieves, Universidad de Granada, 18016 Granada, Spain; 4Department of Surgery, Division of Otolaryngology, Universidad de Granada, 18016 Granada, Spain

**Keywords:** genetic epidemiology, genetic association studies, extreme phenotype, exome sequencing, tinnitus

## Abstract

Exome sequencing has been commonly used to characterize rare diseases by selecting multiplex families or singletons with an extreme phenotype (EP) and searching for rare variants in coding regions. The EP strategy covers both extreme ends of a disease spectrum and it has been also used to investigate the contribution of rare variants to the heritability of complex clinical traits. We conducted a systematic review to find evidence supporting the use of EP strategies in the search for rare variants in genetic studies of complex diseases and highlight the contribution of rare variations to the genetic structure of polygenic conditions. After assessing the quality of the retrieved records, we selected 19 genetic studies considering EPs to demonstrate genetic association. All studies successfully identified several rare or de novo variants, and many novel candidate genes were also identified by selecting an EP. There is enough evidence to support that the EP approach for patients with an early onset of a disease can contribute to the identification of rare variants in candidate genes or pathways involved in complex diseases. EP patients may contribute to a better understanding of the underlying genetic architecture of common heterogeneous disorders such as tinnitus or age-related hearing loss.

## 1. Introduction

A clinical phenotype is a set of observable signs, symptoms, and behavioral features associated with a human disorder. The phenotype includes multiple features or traits and it may be categorical (male or female sex) or quantitative (glucose levels or hearing thresholds). These observable variations in the phenotype of a disorder is known in Mendelian genetics as expressivity and it may range from mild to severe [1,2] Phenotypic variation in quantitative traits can be represented by a bell-shaped graph where mild and severe phenotypes are located at the tails of the distribution. However, the majority of the subjects show an intermediate phenotype (Figure 1).

The genetic architecture of human diseases allows a better understanding of the genetic variants that can influence the phenotype in complex diseases [3]. Next-Generation Sequencing (NGS) technology has been used to uncover missing heritability and elucidate the genetic contribution to common and rare diseases with underlying heterogeneity. In particular, Whole-Exome Sequencing (WES) provides an opportunity to capture rare and ultra-rare alleles of protein-coding genes, which highly influence disease risk. In the last few years, several novel genes have been identified by utilizing WES for various neurological diseases, such as epileptic encephalopathies (*KCNQ2, STXBP1*, and *KCNB1)* and Parkinson’s disease (*VPS13C, ARSB, PTPRH, GPATCH2L*, and *UHRF1BP1L*) [4,5,6].

A significant increase in the prevalence of complex diseases such as bipolar disorder, coronary artery disease [7], type 2 diabetes, hypertension, obesity, and cancer has been reported the last decades [8]. This increase could be related to environmental factors such as diet or lifestyle changes. However, the genetic contribution to complex conditions is still largely unknown, since the contribution of rare variations to heritability is still undetermined. There are several factors that limit the power of gene discovery approaches, such as phenotypic variance [9], the overlap of clinical features observed for similar conditions, minor allelic frequency (MAF), the heterogeneous nature of loci, and the low effect size of potential risk alleles [10].

There is a well-established inverse relationship between the allelic frequency of a given variant and its effect size on the phenotype (Figure 2). The underlying hypothesis is that extreme phenotypes (EP) will occur in extreme cases with an excess of rare variants, with a moderate effect size on the phenotype in addition to the effect of the common variants for the trait of interest. The EP strategy aims to identify rare genetic variants causing a large effect on disease risk [11,12]. The EP study design includes the selection of individuals whose phenotypes are at the extreme ends of a disease phenotype distribution. These extreme subjects may be characterized by early or late age of onset, benign or severe forms of disease, family history, fast progression of symptoms, very high or very low scores in psychometric tests, or extreme levels of a biomarker [13,14,15]. This strategy may identify rare genetic variants by sequencing a relatively small sample size and it can target novel candidate genes, since rare variants that contribute to a particular trait are enriched at the two extremes of a disease distribution [10]. The combination of EP with WES has successfully identified several rare variants and candidate genes for diabetic retinopathy [16], bipolar disorder [17], and cystic fibrosis [18] across diverse ethnic groups.

The aim of this systematic review is to critically analyze the contribution of strategies based on EPs to uncover rare or novel variants or candidate genes in genetic studies of complex disorders.

## 2. Materials and Methods

### 2.1. Study Design

This is a systematic review of genetic studies in complex diseases and it follows Preferred Reporting Items for Systematic Reviews (PRISMA) guidelines (Appendix A) [20] and recommendations from the Human Genome Epidemiology Network (HuGENet) review handbook (https://www.cdc.gov/genomics/hugenet/).

### 2.2. Search Strategies

Literature search for EP strategies was performed on 12 December 2019 using two bibliographic databases (PubMed and Embase). For EP strategies the keywords “phenotypic extreme”, “extreme phenotype”, “rare variant” and “genetics” were used to formulate the search string. The selected keywords could appear in the title, abstract, text word, author keywords, or MeSH Terms of the articles. The keyword string used for the literature search in PubMed was: ((((“phenotypic extreme”[Title/Abstract] OR “extreme phenotype”)[Title/Abstract] AND (“rare variant”[Title/Abstract] OR “genetics”)[Title/Abstract])) OR ((“phenotypic extreme”[Text Word] OR “extreme phenotype”)[Text Word] AND (“rare variant”[Text Word] OR “genetics”)[Text Word])) OR ((“phenotypic extreme” OR “extreme phenotype”) AND (“rare variant” OR “genetics”) [MeSH Terms]); that for Embase was: (‘phenotypic extreme’: ti, ab, kw OR ‘extreme phenotype’: ti, ab, kw) AND (‘rare variant’: ti, ab, kw OR ‘genetics’: ti, ab, kw) AND [2009–2019]/py AND [english]/lim. Records published in the last 10 years, studies in English language, and only human studies were included in the literature search by configuring filters if available, e.g., on PubMed.

### 2.3. Research Question and Selection Criteria

The objective of this systematic review is to assess the evidence supporting the design of genetic studies using extreme phenotype strategies to find rare or novel variants or genes involved in complex disorders. According to this hypothesis, we formulated the following research question: “Are EP strategies useful to establish the genetic contribution in complex diseases?”. To answer this question, we followed the “Population, Intervention, Comparison, Outcome, Study design” (PICOS) strategy:Population: Patients with a complex disease or condition.Intervention: Selection of individuals according to extreme phenotype criteria (i.e., early onset, fast progression of disease, very high or very low scores in psychometric tests, or extreme levels of a biomarker).Comparison: Genetic association studies (genome-wide association studies (GWAS), WES, genotyping, Sanger sequencing, or targeted sequencing).Outcome: genetic findings reported (rare variants, candidate genes, or pathways associated with the condition of interest).Study design: case–control, case report, case–cohort, or trios.

### 2.4. Exclusion Criteria

Studies in non-human models.Studies not published in English.Studies with a publication date ≥10 years.

### 2.5. Quality Assessment of Selected Studies

The extracted records were screened to remove duplicate entries. The title and abstract of all articles were reviewed to exclude reviews, meta-analysis, and irrelevant records (non-genetic studies, pharmacogenomics or clinical studies). The search was conducted primarily for rare variants, but any type of variants were retained and included in this systematic review. After screening, the obtained records were considered for full-text assessment in the next step. To assess the quality of these articles, we formulated 8 questions for EP studies (Table 1). For each question, a positive answer was scored as 1 and a negative answer as 0. Each author classified and rated each record independently of each other. Differences in the scores were discussed to get a final consensus score. If a record achieved ≥60% of the total score, the response to Q8 was “yes”, and the reported rare variants have a MAF < 0.05, then the record was selected for synthesis. So, only studies with significant results were included. Two of the authors carried out the synthesis (SA, JALE). The outcome for each selected study was assessed according to Q8 and the following criteria: if a given study had found any rare or de novo variant, common variant, copy number variants, candidate genes, or pathways for EP subjects, then the major outcome was considered as positive.

### 2.6. Data Extraction and Synthesis

The following information was extracted from each article selected for synthesis: first author’s last name, publication year, disease/disorder name, population ancestry, study design, sequencing method, EP/disease phenotype criteria, sample size for cases, age of disease onset, sex of individuals, MAF, and main genetic findings. Moreover, the phenotype criteria and the main genetic findings for EP were of great interest for synthesis.

### 2.7. Risk of Bias

The Cochrane collaboration tool [21] was used to assess the risk of bias for each selected study (Appendix A).

## 3. Results

### 3.1. Selection and Characteristics of EP Studies

For the EP strategy, we retrieved 106 records in total, 66 records from PubMed and 40 from Embase, by using the search strings reported in the search strategy section. After duplicates’ removal, we retained 89/106 records aggregated from the two databases. Next, after screening by title and abstract of the articles, we retrieved 30/89 records that were included for full-text assessment. The discarded records were reviews, meta-analyses, non-genetic studies, pharmacogenomics studies, posters, or abstracts presented at scientific meetings. All studies including variants with MAF > 0.05, single cases, or <5 patients with EP were excluded. We performed quality assessment for 30 articles, and 19/30 records surpassed the minimum quality assessment score and were considered for synthesis. (Figure 3, Appendix A).

Among the 19 studies selected for synthesis, 16 records were related to physical conditions, 1 was on bipolar disorder, and 2 were related to neurological disorders including epilepsy and Alzheimer’s disease. All of these studies reported rare variants, candidate genes, or potential pathways associated with a particular trait using an EP approach. These 19 EP studies covered 18 complex diseases.

Information about population ancestry and sample size of cases was available for all 19 studies. Only 11/19 studies reported the age of disease onset, and 18/19 records reported the sex of the individuals. The most common criteria to define EP included early onset, late onset, family history, acute form, and/or fast progression of a disease. In addition, disease-specific features were also considered to define an EP, such as the worst score in biomarkers levels including Bone Mass Density (BMD) and spirometry-based severity according to Global Initiative for Chronic Obstructive Lung Disease (GOLD) grade. The reported sample size was between 12 and 32,965 individuals. A summary of the characteristics of these 19 EP studies is shown in Table 2.

### 3.2. Synthesized Findings of EP Studies

In the 19 EP studies, the combination of general and disease-specific EP criteria was used to select individuals. Information on the study design, sequencing technique, and ancestry population was available for all 19 studies. The reported sample size varied according to the design and sequencing method: 1711 ± 2513 (mean ± SD) for GWAS, 929 ± 2389 for genotyping, 1274 ± 9380 for WES, 29 ± 9 for targeted sequencing, and 949 ± 8742 for Sanger sequencing. All 19 examined studies using EP to select individuals reported significant findings including several rare variants, copy number variants, potential candidate genes or pathways associated with the condition of interest. WES was able to find rare variants in 13/19 studies (MAF = 0.00–0.05) in identified variants. It also helped in the identification of several novel candidate genes including *TACC2* [22], *PRKCD, C1QTNF4, DNMT3A* [23], *LOC728699*, and *FASTK* [16]. GWAS identified a rare variant in 1/19 study (MAF = 0.04). In addition, genotyping and targeted and Sanger sequencing contributed in the identification of many candidate genes and micro-deletions.

## 4. Discussion

### 4.1. Summary of the Main Findings

Our systematic review shows that individuals with an EP may reveal rare variants that can influence genetic susceptibility in most complex disorders. Complex disorders have a heterogeneous spectrum of symptoms, with variable expressivity observed in each patient. By cluster analysis, it is possible to identify subgroups of patients, and by selecting patients with EP (high expressivity), we would expect to find an enrichment of rare variations associated with the EP [37]. However, we cannot recommend a particular EP strategy to select patients, although the selection of individuals with an early-onset disease and/or a severe phenotype (genetic anticipation) will probably help in the search of rare variations. In contrast, elderly patients can show mutations associated with exposure to environmental factors along life (ultraviolet radiation, chemical agents, pollutants) [38]. In general, the criteria to define EP combine common and disease-specific features such as the chronic state of a disease, very high or low biomarker levels such as BMD, spirometry-based severity level according to GOLD, family history, and early/late age of disease onset.

Of note, a large sample size was not required in WES studies for the discovery cohort, and 10/19 records had a number of cases <100. Therefore, a moderate sample size of individuals with EP was sufficient to identify candidate rare variants or genes. These individuals with EP were carriers of rare variants with a high effect size to target new candidate genes. The EP approach was reproducible across different populations, since the selected studies recruited cases with different ethnic backgrounds including Asian, African, and European ancestry and with monogenic diseases such as cystic fibrosis [13] with an extreme phenotype (persistent tracheobronchial infection with early onset) [39]. Therefore, the information about age of disease onset and sex of the selected individuals is essential to define an EP [40].

### 4.2. Selection of EP in Quantitative Traits

Individuals with EP are characterized by extreme clinically relevant attributes, toxic effects, or extreme responses to a treatment [1]. From a theoretical perspective, a very EP is more informative than an almost EP, but in practice there are several limitations associated with the very EP, such as vulnerability to phenotype heterogeneity and measurement errors. If a significant proportion from both sides of an extreme is discarded, the almost EP can still be more powerful than random sampling of the same size. The benefits of EP sampling were demonstrated by proposing power calculation methods with the help of the maximum likelihood approach [11,41]. It was also indicated that EP sampling to detect rare variants is more cost-efficient as compared to traditional study designs with a large cohort [42]. Replication in a second independent EP cohort to enhance the power of a study is highly recommended, but it is unlikely to obtain a large sample size of EP subjects from a single region [43]. However, the EP approach is considered more efficient than random sampling for the detection of rare variants associated with a trait [11]

### 4.3. Familial Disorders and EP Strategy

Some common disorders show rare familial phenotypes with Mendelian inheritance associated with rare variants with large effect size. There are many studies using the EP strategy for familial cases of complex disorders, such as Alzheimer’s disease (AD) [25], polyautoimmunity disorder [24], and congenital hypothyroidism [44]. For example, a recent study using linkage analysis demonstrated that by selecting individuals with familial autoimmunity and polyautoimmunity as EP, it was possible to identify the *SRA1* gene (LOD score = 5.48) [24]. Furthermore, a WES study on AD analyzed non-Hispanic White patients and Caribbean Hispanic families to find genes associated with early-onset AD. Heterozygous non-synonymous variants with global MAF < 0.001 were selected for variant prioritization and showed autosomal-dominant segregation in these families. Several genes such as *RUFY, TCIRG1, PSD2*, and *RIN3* were identified that could be involved in endolysosomal transport in both early- and late-onset AD [25]. In some complex diseases such as Meniere disease (MD), a syndrome characterized by hearing loss, episodic vertigo, and tinnitus, there is also a strong evidence of genetic predisposition in most affected families, showing an autosomal-dominant inheritance with almost 60% penetrance. By using WES in familial MD analysis, a burden of multiplex rare missense variants in the *OTOG* gene was reported in 30% of familial cases [45], which illustrates the success of considering familial cases as EP. Furthermore, a study on genetic epilepsy with hay febrile seizures plus (Dravet syndrome) has reported a *SCN1A* missense variant in a large Jewish family (14/17 cases) with epilepsy syndrome at both extremes (low and high) [15], and a study on thyroid dysgenesis with congenital hypothyroidism found a familial *PAX8* variant associated with EP [44].

### 4.4. An EP Strategy to Investigate the Genetic Contribution to Tinnitus

Tinnitus is the perception of noise in the absence of an external acoustic stimulation, affecting more than 15% of the population and causing a decrease in health-related quality of life [46]. Several specific instruments have been defined to characterize chronic or severe tinnitus, and these instruments have been proposed to measure tinnitus annoyance to define EP for genetic studies [47]. Epidemiological evidence to support a genetic contribution to tinnitus is still weak because of the heterogeneous nature of this condition. In fact, tinnitus can occur together with multiple comorbidities including hearing loss, migraine, sleep disorders, anxiety, other psychological conditions, and some rare monogenic disorders [48]. The careful selection of phenotypes for genetic studies is crucial. The inclusion criteria should consider young individuals with severe forms of bilateral tinnitus to investigate the genetic contribution of rare variations to tinnitus. These individuals may carry a greater susceptibility and lower environmental load; however, severe forms of tinnitus in young individuals are rare [49] and multicenter studies are needed to reach a minimum sample size [50]

### 4.5. Limitations

Some weaknesses were found in the design of EP strategies; therefore, further research is required. The replication of the genetic studies across different populations with different ethnic backgrounds has enough potential to validate genetic associations [13,36]; however, the frequency of allelic variants is different across different populations, and specific reference data for allelic frequencies are needed for each population. The rare variants reported in simplex families with EPs should be validated in more patients with a severe phenotype [24]. Most of the studies used WES rather whole-genome sequencing (WGS) and this can cause the loss of useful genetic information and erroneous results in calculating the effect size of rare variants at the individual level across a particular phenotype [17].

## 5. Conclusions

Genetic studies have confirmed the effectiveness of the EP strategies to establish the genetic contribution of rare variations to complex diseases.

## Figures and Tables

**Figure 1 genes-11-00987-f001:**
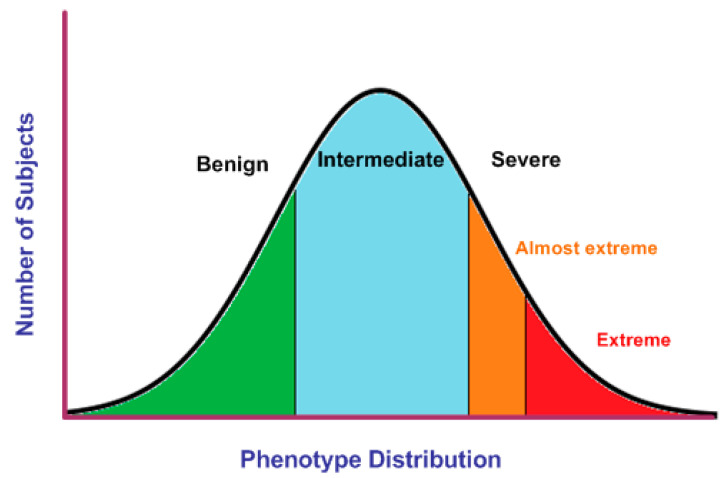
Phenotypic variation in quantitative traits. Individuals’ phenotypes can be classified as benign, intermediate, or severe according to general and disease-specific criteria. Extreme phenotypes are identified at the ends of the normal distribution (green, orange, and red areas).

**Figure 2 genes-11-00987-f002:**
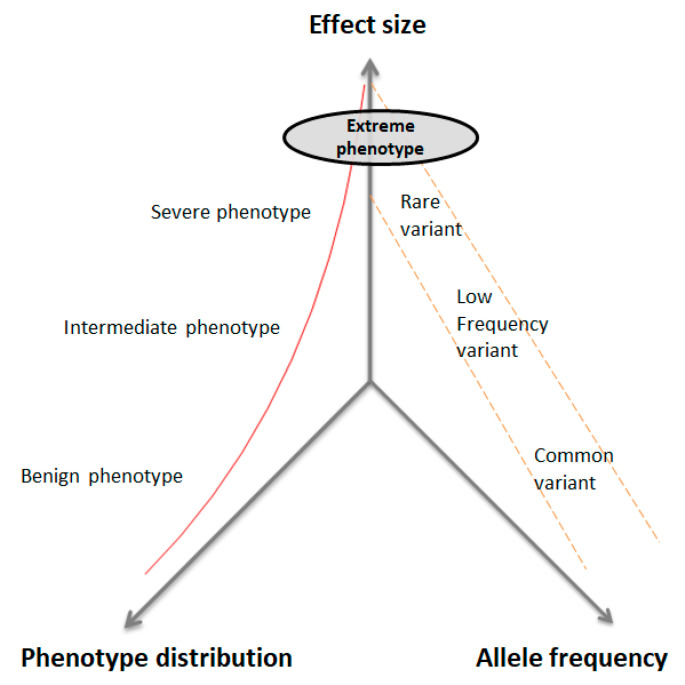
Distribution of genetic variants according to allelic frequency and effect size on the phenotype in quantitative traits. Individuals with extreme phenotypes will show a burden of rare variations with a moderate to large effect size (modified from Manolio et al., 2008 [19]).

**Figure 3 genes-11-00987-f003:**
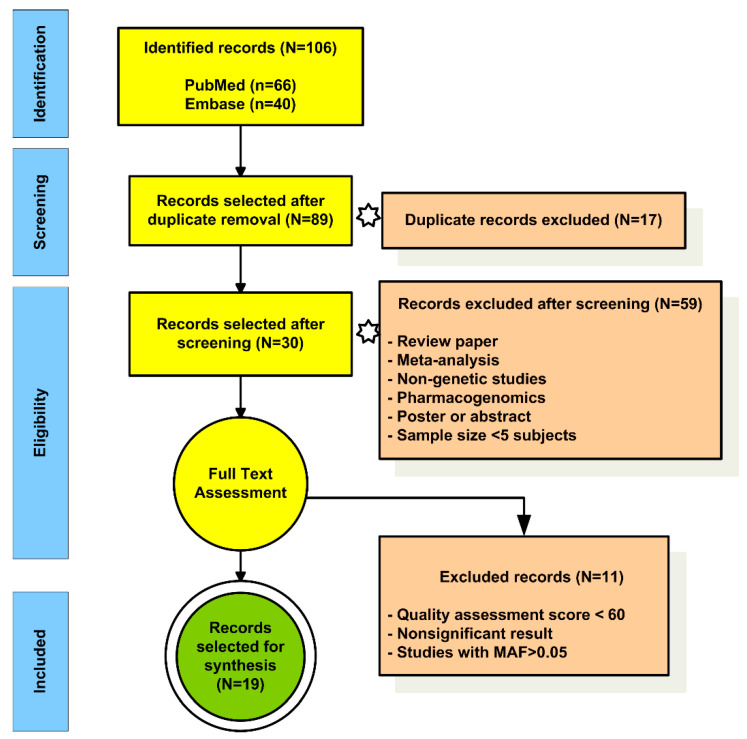
Flowchart to select extreme phenotype records for synthesis.

**Table 1 genes-11-00987-t001:** Criteria used to assess the quality of the selected genetic studies using an extreme phenotype approach.

No.	Question	Answer
Q1	Is there a thorough description of the study design?	Yes/No
Q2	Has the study described the method of sequencing/genotyping?	Yes/No
Q3	Has the study provided information about population ancestry?	Yes/No
Q4	Is there any information on the sex of the selected individuals?	Yes/No
Q5	Is there any information on the age of disease onset?	Yes/No
Q6	Has the study used extreme phenotype criteria for sample recruitment?	Yes/No
Q7	Has the study performed sex-specific analysis for genetic associations?	Yes/No
Q8	Has the study reported significant genetic findings?	Yes/No

**Table 2 genes-11-00987-t002:** Summary of the 19 genetic studies using an extreme phenotype approach selected for synthesis.

Reference	Disease	EP Criteria	Study Design	Sequencing Method	Ancestry	Number of Patients	Onset	Sex	Genetic findings		AF (Ancestry-Dependent)
									Gene/ Pathway	SNP/Indel	
Pullabhatla et al. (2017) [23]	Systemic lupus erythematosus	Proband with early onset and clinical features with poor outcome	Family trios, Replication cohort	WES	EU	30 trios, 10995	<25 y	Not reported	*PRKCD*	3: 53223122 G>A	De novo variants and novel genes
									*C1QTNF4*	11: 47611769 G> C	
									*DNMT3A*	2: 25457236 G> A	
Johar et al. (2016) [24]	Polyautoimmunity	Polyautoimmunity and familial autoimmunity	Case–control, Cross-sectional	WES	Colombian	47	Not reported	M,F	*PLAUR*	rs4760	0.1
									*DHX34*	rs151213663	0.004
									*SRA1*	rs5871740, rs202193903	Not found
									*ABCB8*	7:150744528:G>T, 7:150744370: CGT/-	Not found
									*MLL4*	rs186268702	0.0007
Kunkle et al. (2017) [25]	Alzheimer’s disease	Early-onset Alzheimer’s disease, familial or sporadic	Case–control, Replication cohort	WES	NHW and Caribbean Hispanic	93, 8570	<65 y	M,F	*RUFY1*	5:179036506:T>G	0.001
									*RIN3*	14:93022240:G>T	0.0005
									*TCIRG*	11:67810477:C>T	0.0007
									*PSD2*	5:139216541:G>A, 5:139216759:G>A	0.0006, 0.00005
Emond et al. (2012) [13]	Cystic fibrosis (CF)	CF with early onset of persistent *Pseudomonas aeruginosa* infection	Case–control, Replication cohort	WES	EU America, African American, White Hispanic, NHW, Asian, Aleut	43, 696	≤2.5 y	M,F	*DCTN4*	rs11954652, rs35772018	0.048, 0.017
Shtir et al. (2016) [16]	Diabetes	Diabetes for at least 10 years without diabetic retinopathy	Case–control, Cross-sectional	WES	Saudi	43	Not reported	M,F	*FASTK*	7:150774771:C>T, 7:150777859:A>T	0, 0
									*LOC728699*	rs149540491, rs117616768, 12:20704520:C>A	0.05, 0.01, 0.02
Liu et al. (2016) [26]	Lung cancer	Familial or sporadic lung cancer cases, ever smokers or severe chronic obstructive pulmonary disease (COPD)	Case–control, Cross-sectional	WES	NHW	48 sporadic 54 familial	56 y familial 61 y sporadic	M,F	*DBH*	rs76856960	0.0034
									*CCDC147*	rs41291850	0.0026
Husson et al. (2018) [17]	Bipolar I disorder	Family history of mood disorder and early onset	Case–control, Cross-sectional	WES	EU	92	mean: 24 y	M,F	>13 genes	>100 SNPs	0.000015-0.009
Johar et al. (2015) [27]	Multiple autoimmune syndrome	Multiple autoimmune syndrome with Sjögren’s syndrome	Case–control, Cross-sectional	WES	Colombian	12	28–67 y	F	*LRP1/STAT6*	12:57522754:A>C	Novel mutation
Hiekkala et al. (2018) [28]	Hemiplegic migraine	≥2 migraine attacks, completely reversible motor weakness	Case report, Cross-sectional	WES	Finnish	293	median: 12 y	M,F	*ATP1A2*	rs765909830, 1:160100376:G>A	0, 0
									*CACNA1A*	rs121908212	0
Qiao et al. (2018) [29]	COPD	COPD cases with GOLD grade 3 or 4	Case–control, Cross-sectional	WES	EU, NHW, African American	≈1769	>45 y, ≤65 y	M/F	jak-stat signaling pathway	-	Not reported
									*TBC1D10A, RFPL1*	Not reported	
Bruse et al. (2016) [22]	COPD	COPD cases with GOLD grade 3 or 4	Case–control, Cross-sectional	WES	NHW	62	Not reported	M/F	*TACC2*	chr10:123842508, 10:123844900, 10:123903149, 10:123970638, 10:123987443, 10:123996970, 10:124009124	0.000008901, 0.000008796, 0.001851, 0.000008999, Not found 0.03476, 0.07
Nuytemans et al. (2018) [30]	Thrombotic storm (TS)	Severe onset of ≥2 arterial, unusual clot location, refractory, reoccurrence	Case report, Cross-sectional	WES, Targeted sequencing	White and Indian	26 (13 trios)	Not reported	M,F	*STAB2*	rs779748342, rs758868186, rs201799617, rs17034336, rs149382223	Not found, Not found, 0.0002, 0.0441, 0.0008
									*CHPF*	2:220405189:C>T	Not found
									*CHST3*	rs145384892	Not found
									*SLC26A2*	rs104893919, rs78676079	Not found, 0.0076
									*CHST12*	rs17132399	Not found
									*CHPF2*	rs776052782, rs117332591, rs377232422	Not found, 0.0028, Not found
									*CHST15*	rs34639461	0.011
									*PAPSS2*	rs45467596	0.0219
Aubart et al. (2018) [31]	Marfan syndrome	Severe aortic features (dissection or preventive thoracic aortic aneurysm rupture surgery at a young age) or sib pairs	Case–control, Cross-sectional	WES	EU	51 EP and 8 sib-pairs	≈10–30 y	M,F	*COL4A1*	c.4615C>T, c.1630G>C, c.4453T>C,	0.02, 0.04, 0.003
									*FBN1*	c.1585C>T	0
									*SMAD3*	c.6424T>C	0
Gregson et al. (2018) [32]	Bone mass density	Extremely high or moderately high bone mass density	Case–control, Replication cohort	GWAS	EU	1258, 32965	Not reported	M,F	*WNT4/ZBTB40*	rs113784679	0.04
Lee et al. (2018) [33]	Ulcerative colitis	Ulcerative colitis patients with good or poor prognosis	Case–control, Replication cohort	Genotyping	Korean	881, 274	35.6 ± 13.9 y	M,F	*HLA-DRA* and *HLA-DRB*	rs9268877	0.000
Tomaiuolo et al. (2012) [34]	Acute myocardial infarction (AMI)	AMI patients with first episode before or after 45 years of age	Case–control, Replication cohort	Genotyping	EU	1653, 909	Not reported	M,F	*MTHFR C677T, FII G20210A, Factor V Leiden*	-455G>A	-
Goldberg-Stern et al. (2013) [15]	Epilepsy with febrile seizures plus	Generalized epilepsy with febrile seizures plus, a proband with Dravet syndrome	Case-control, Cross-sectional	Sanger sequencing	Ashkenazi Jewish	14 familial cases	infancy to 7 y	M,F	*SCN1A*	c.4114A>G: p.K1372E; exon 21	-
Shen et al. (2017) [35]	Spermatogenic failure	Spermatogenic failure with azoospermia, mild oligozoospermia or severe oligozoospermia	Case–control, Cross sectional	Sanger sequencing	Chinese Han	884	Not reported	M	*MAGEA9*	Deletion (chrX:149580739-149580850)	-
Uzun et al. (2016) [36]	Preterm birth	Patients delivering <34 weeks	Case report, Cross-sectional	Targeted Sequencing of 329 genes	African-American; Asian; Hispanic; White; Native American	32	Not reported	F	*WASF3*	rs17084492	0.01357(NFE), 0.07(African)
									*AZU1*	rs28626600	0.1(NFE), 0.01662(African)

Legend: NHW, Non-Hispanic White; EU, European; WES, Whole-Exome Sequencing; GWAS, genome-wide association studies; EP, extreme phenotype; SNP, Single Nucleotide Polymorphism; AF, allelic frequency.

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
