# Peer review of "A Systematic Review of Extreme Phenotype Strategies to Search for Rare Variants in Genetic Studies of Complex Disorders"

_genes, 2020, doi:10.3390/genes11090987_

Round 1

Reviewer 1 Report

This paper investigates the utility of the extreme phenotyping selection strategy in finding novel rare variants associated to complex phenotypes. Through a systematic search in the literature the authors report 19 publications for which this strategy successfully uncovered new variants.

Comments:

By using as a criterion publications reporting only significant associations (Q8), claiming that this approach is useful can only be biased. To their defense, there are probably not many "negative" studies published in the literature, and authors only expose the publication bias towards significant results. Maybe the authors should mention it as a limitation of their study.

Minor corrections:

Line 21: I suggest "polygenic" instead of multiallelic. Multiallelic is usually a term applied to a locus.

Line 43: "... important role to ..."

Lines 59-61: This sentence is not clear. Do the authors mean that rare variants will bring an additional contribution in top of the common ones already found? Please reformulate.

Line 97: PICOS instead of PICO.

Line 107: In the exclusion criteria, add "publications with publication date >= 10 years", although all exclusions were already mentioned before in Lines 90-91.

Header of Table 2: What means "Not sex"?

Table 2: In some instances, the reported allele frequency is 0. What does this mean? Is it rounded?

Support Table S2: I'm not sure that biasness is a real word. I suggest to simply use bias. 

Author Response

Response to Reviewer #1:

Comments:

By using as a criterion publications reporting only significant associations (Q8), claiming that this approach is useful can only be biased. To their defense, there are probably not many "negative" studies published in the literature, and authors only expose the publication bias towards significant results. Maybe the authors should mention it as a limitation of their study.

We thank the reviewer the useful comments to improve our study. Among the 30 EP studies selected, only 2 of them did not find significant findings (shown in Table S3). Although this could be a publication bias, these studies do not provide information about population ancestry nor perform a sex specific analysis and they were not considered because of these criteria.

We have revised all minor comments in the text.

Minor corrections:

Line 21: I suggest "polygenic" instead of multiallelic. Multiallelic is usually a term applied to a locus.

We have changed the text on Line 21.

Line 43: "... important role to ..."

We have modified the text on Line 43.

Lines 59-61: This sentence is not clear. Do the authors mean that rare variants will bring an additional contribution in top of the common ones already found? Please reformulate.

We have modified this sentence on Line 63.

Line 97: PICOS instead of PICO.

We have revised on Line 107.

Line 107: In the exclusion criteria, add "publications with publication date >= 10 years", although all exclusions were already mentioned before in Lines 90-91.

We have added the text on Line 120

Header of Table 2: What means "Not sex"?

 It is a typo mistake, we have corrected to sex.

Table 2: In some instances, the reported allele frequency is 0. What does this mean? Is it rounded?

The variants with AF=0 were not reported in the reference population or control group. They were not rounded.

Support Table S2: I'm not sure that biasness is a real word. I suggest to simply use bias. 

We have corrected the text to bias in Support Table S2.

Reviewer 2 Report

The authors have reviewed current literature on using selected individuals with extreme  phenotypes to establish the genetic contribution to complex diseases. This is a useful piece of work for the complex disease and rare disease field. Overall I would like more clarity over the different types of study and analysis that this systematic review identified. I would suggest that using families to identify monogenic disease causing variants is not a EP strategy.  This may require some reorganisation of the text and additions to table 2.

Major Comments

Without performing a full systematic review, it appears that there are two different types of studies identified using the review terms and this needs to be clarified:

1 – sequencing of likely monogenic disease patients and their families to identify monogenic cause of their disease

2 – Association studies of individuals using EP as case control status for association analysis

These could be dealt with independently in this manuscript or the focus should be on the second.

The work as performed and described uses the terms ‘complex diseases’ and ‘rare variants’. In this reviewers experience, the term ‘rare variants’ is used to describe variants associated with rare monogenic disease and not as the opposite of common variants. It is unlikely that the hidden heritability of complex diseases will be fully explained by rare variants (my definition would be frequency of less than 1 allele in 10,000) and much more likely that numerous ‘low frequency’ variants will be identified with medium effect sizes. Any truly rare variants (including those that are denovo) will likely represent known or unknown monogenic disorders that are phenotypically similar to complex disorders – as discussed on line 215.  To this end, the review criteria and the language used in the manuscript could be better justified to the reader.

It would be useful to better understand the type of analysis used for each publication. Perhaps add to table 2. The methods are very different from identification and manual curation and filtering of denovo variants in family’s to using single locus association tests in case control cohorts.

Within the selection criteria I would like to a criteria referring to power. Many of the studies selected are underpowered and low frequency/rare variants have been identified by chance.  This needs to be acknowledged and this in itself is strength of the EP strategy when considering costs.

The fundamental reason for performing EP experiments is not discussed – this is largely due to costs. An ideal experiment would be to genotype/sequence everyone who you have phenotyped.  I think this should be mentioned.

Whole genome genotyping techniques use genome wide SNP arrays, corresponding genotype calling and imputation for GWAS are not designed to genotype and detect low frequency or rare variants. This information should be added as a for the Gregson paper in Table 2.

Minor Comments

A comprehensive  proof read is required. Some references are listed twice (Goldberg- Stern).

The word ‘variant’ should be used instead of ‘mutation’ in all parts of this manuscript that refer to inherited or denovo variation (i.e. not somatic).

Line 35 - Expressivity in mendelian genetics refers to the degree to which a phenotype differs in individuals with the same monogenic variant due to environment and / or genetic ‘background’ .

Anticipation is also a very specific term usually used to describe how the phenotype of a specific disease caused by a specific variant in a specific family worsens over generations, usually due to the increasing expansion of a trinucleotide repeat.

Paragraph starting line 52- I would suggest that a figure showing allele frequency versus effect size could also be useful in introduction.

Author Response

Response to Reviewer #2:

The authors have reviewed current literature on using selected individuals with extreme phenotypes to establish the genetic contribution to complex diseases. This is a useful piece of work for the complex disease and rare disease field. Overall I would like more clarity over the different types of study and analysis that this systematic review identified. I would suggest that using families to identify monogenic disease causing variants is not a EP strategy.  This may require some reorganisation of the text and additions to table 2.

We thank the reviewer the useful comments to improve our study. We have revised the text according to reviewers’ suggestions and included a new figure 2 in the introduction on Page 3 and Lines 61-62.

Major Comments

Without performing a full systematic review, it appears that there are two different types of studies identified using the review terms and this needs to be clarified:

1 – sequencing of likely monogenic disease patients and their families to identify monogenic cause of their disease

2 – Association studies of individuals using EP as case control status for association analysis

These could be dealt with independently in this manuscript or the focus should be on the second.

We agree with this comment. Among the 19 genetic studies selected for synthesis, some of them included familial monogenic variants of some complex disorders (SLE, familial Alzheimer); however, the search strategy was designed to find studies that used EP to search for rare variants, regardless of the complex disease has familial monogenic endophenotypes.

The work as performed and described uses the terms ‘complex diseases’ and ‘rare variants’. In this reviewers experience, the term ‘rare variants’ is used to describe variants associated with rare monogenic disease and not as the opposite of common variants. It is unlikely that the hidden heritability of complex diseases will be fully explained by rare variants (my definition would be frequency of less than 1 allele in 10,000) and much more likely that numerous ‘low frequency’ variants will be identified with medium effect sizes. Any truly rare variants (including those that are denovo) will likely represent known or unknown monogenic disorders that are phenotypically similar to complex disorders – as discussed on line 215.  To this end, the review criteria and the language used in the manuscript could be better justified to the reader.

It would be useful to better understand the type of analysis used for each publication. Perhaps add to table 2. The methods are very different from identification and manual curation and filtering of de novo variants in family’s to using single locus association tests in case control cohorts.

We agree with this comment and linkage with segregation analysis performed in families is different from genetic association studies using a case-control design. All the selected studies in Table 2 used a case-control design, being the EP the case group.

Within the selection criteria I would like to a criteria referring to power. Many of the studies selected are underpowered and low frequency/rare variants have been identified by chance.  This needs to be acknowledged and this in itself is strength of the EP strategy when considering costs.

We have detailed the factors that influence the statistical power in lines 56-59: “There are several factors that limit the power of gene-discovery approaches such as phenotypic variance, the overlap of clinical features with similar conditions, the minor allelic frequency (MAF), heterogeneous nature of loci, and the low effect size of the potential risk alleles.” The first criteria used was “the description of the study design”, including the statistical methods. Most of the selected studies used a large number of individuals in the control group and some of them included a replication cohort to validate the association of rare variants with EP.

The fundamental reason for performing EP experiments is not discussed – this is largely due to costs. An ideal experiment would be to genotype/sequence everyone who you have phenotyped. I think this should be mentioned.

We also agree that the main benefit of EP strategy is the cost saving. We have mentioned this in the discussion on line 221:  “…EP sampling to detect rare variants is more cost efficient”.

Whole genome genotyping techniques use genome wide SNP arrays, corresponding genotype calling and imputation for GWAS are not designed to genotype and detect low frequency or rare variants. This information should be added as a for the Gregson paper in Table 2.

We fully agree with this comment, GWAS using SNP array are not able to find rare variants and they are currently replaced by sequencing technology.

Minor Comments

A comprehensive proof read is required. Some references are listed twice (Goldberg- Stern).

We have removed the duplication.

The word ‘variant’ should be used instead of ‘mutation’ in all parts of this manuscript that refer to inherited or denovo variation (i.e. not somatic).

We have revised it in the text.

Line 35 - Expressivity in mendelian genetics refers to the degree to which a phenotype differs in individuals with the same monogenic variant due to environment and / or genetic ‘background’.

We have modified the sentence in line 35.

Anticipation is also a very specific term usually used to describe how the phenotype of a specific disease caused by a specific variant in a specific family worsens over generations, usually due to the increasing expansion of a trinucleotide repeat.

Paragraph starting line 52- I would suggest that a figure showing allele frequency versus effect size could also be useful in introduction.

We have modified the introduction and added a new figure 2 on Page 3.